# Histologic Grade Is Predictive of Incidence of Epidermal Growth Factor Receptor Mutations in Metastatic Lung Adenocarcinoma

**DOI:** 10.3390/medsci5040034

**Published:** 2017-12-11

**Authors:** Michelle Levy, Liisa Lyon, Erika Barbero, John Wong, Jennifer Marie Suga, Danny Sam, Minggui Pan

**Affiliations:** 1Internal Medicine Residency Program, Kaiser Permanente, Santa Clara, CA 95051, USA; mlevy1@pennstatehealth.psu.edu (M.L.); Erika.m.barbero@kp.org (E.B.); jowong@llu.edu (J.W.); Danny.sam@kp.org (D.S.); 2Division of Research, Kaiser Permanente, 2000 Broadway, Oakland, CA 94612, USA; liisa.l.lyon@kp.org; 3Division of Oncology and Hematology, Kaiser Permanente, Vallejo, CA 94589, USA; jennifer.m.suga@kp.org; 4Department of Oncology and Hematology, Kaiser Permanente, Santa Clara, CA 95051, USA

**Keywords:** EGFR mutation, lung adenocarcinoma, histologic grade, NSCLC, erlotinib

## Abstract

Activating epidermal growth factor receptor (EGFR) mutations in metastatic non-small cell lung cancer (NSCLC) are associated with a high response rate to EGFR tyrosine kinase inhibitor (TKI). The current guidelines recommend routine EGFR mutational analysis prior to initiating first line systemic therapy. The clinical characteristics including smoking status, histologic type, sex and ethnicity are known to be associated with the incidence of EGFR mutations. We retrospectively analyzed 277 patients with metastatic NSCLC within Kaiser Permanente Northern California (KPNC); among these patients, 83 were positive for EGFR mutations. We performed both univariate and multivariable logistic regressions to identify predictors of EGFR mutations. We found that histologic grade was significantly associated with the incidence of EGFR mutation, regardless of ethnicity, sex and smoking status. In grade I (well differentiated) and II (moderately differentiated), histology was associated with significantly higher incidence of EGFR mutations compared to grade II–III (moderate-to-poorly differentiated) and III (poorly differentiated). Ever-smokers with grade III lung adenocarcinoma had 1.8% incidence of EGFR mutations. This study indicates that histologic grade is a predictive factor for the incidence of EGFR mutations and suggests that for patients with grade II–III or III lung adenocarcinoma, prompt initiation of first-line chemotherapy or immunotherapy is appropriate while awaiting results of EGFR mutational analysis, particularly for patients with history of smoking.

## 1. Introduction

The discovery of classic epidermal growth factor receptor (EGFR)-activating mutations more than a decade ago changed the treatment landscape for patients with metastatic non-small cell lung cancer (NSCLC) [1,2]. In North America, approximately 10–15% of cases of metastatic NSCLC harbor the classic EGFR-activating mutation [3,4,5]. Four clinical features are associated with higher probability of EGFR mutation: never-smoker, adenocarcinoma, female and Asian descent [3,6,7,8]. Erlotinib, afatinib and gefitinib are EGFR tyrosine kinase inhibitors (TKIs), approved as first line therapy for mutation-positive patients [9,10,11]. 

The current National Comprehensive Cancer Center Network (NCCN) guidelines recommend routine EGFR mutational analysis to determine whether a patient should be given a TKI or systemic chemotherapy as first line therapy [12]. However, the wait-time for the result of EGFR mutational analysis may be significant especially for those patients who are symptomatic with high tumor burden and require timely initiation of systemic therapy. A 2016 study showed that time-to-treatment (TTT) from first medical oncology visit was 34 days, improved to 22 days even after reflex testing of EGFR mutational analysis was implemented [13]. A study from Canada showed that median TTT from the diagnosis of lung cancer to the initiation of treatment was 32 days and that the patients whose treatment was not initiated within 30 days had significantly worse survival compared to the patients whose treatment was initiated within 30 days [14]. Estimating the incidence of EGFR mutation using existing clinical characteristics could be valuable in facilitating decision making as to whether systemic chemotherapy or TKI should be started while awaiting the result of EGFR mutation analysis, hence shortening the TTT, and improving patient outcome. In this study, we aimed to determine if histologic grade could be a valuable clinical factor for estimating the incidence of EGFR mutation in addition to the previously established clinical characteristics, hence guiding clinical decision making.

## 2. Materials and Methods 

### 2.1. Study Population 

This retrospective cohort study was conducted at Kaiser Permanente Northern California (KPNC), a large integrated health care delivery system caring for more than 4 million members who are broadly representative of the local and statewide population, and approved by the Kaiser Permanente Internal Review Board (CN12-MPan-01-H). We identified the KPNC members diagnosed with relapsed or metastatic non-small cell lung cancer (NSCLC) from a laboratory database of the patients whose tumor was tested for EGFR mutation between 2004 and 2012. The EGFR mutational analysis was performed by either Quest Diagnostics (Madison, NJ, USA) or Mayo Clinic Laboratory (Rochester, MN, USA) on a core biopsy, a lobectomy specimen, or in a few cases on cytology block prepared from malignant pleural fluid. The histologic type and grade were retrieved by reviewing the pathology report in the electronic health record (EHR). The pathology reporting was based on the 2004 World Health Organization (WHO) of lung tumors and the American Joint Committee on Cancer (AJCC) version 6 (approved by the College of American Pathologists, CAP) [15,16]. The grading criteria was mainly based on the architecture and cytologic atypia. Low grade (grade I or well-differentiated) was defined as predominantly non-solid architecture with mild cytologic atypia, intermediate grade (grade II or moderately differentiated) was defined as mixed solid pattern and non-solid pattern with moderate cytologic atypia, and high grade (grade III or poorly differentiated) was defined predominantly solid pattern with severe atypia. The moderately to poorly differentiated (grade II to III) was defined as more than half of solid pattern but less than being predominant (less than 90%) and with severe cytologic atypia. Gender, age, race, membership status, date of death, and erlotinib prescriptions were retrieved from the EHR. Erlotinib use was defined as having at least one prescription after diagnosis. The date of study entry was the date of metastatic or relapsed NSCLC definitively confirmed by an imaging study, biopsy, or cytology from pleural effusion. Patients were followed until death or leaving plan membership permanently with censoring at 36-month post-diagnosis. 

### 2.2. Statistical Methods

Demographic and clinical characteristics were tested for association with the EGFR mutation. Categorical variables were evaluated using frequencies and proportions, and associations were tested with Chi-Square and Fisher’s exact test. Medians with 1st and 3rd quartiles [Q1, Q3] were used to describe continuous but non-normally distributed data, and Mann-Whitney U-tests were used for comparisons. We performed both univariate and multivariable logistic regressions to identify predictors of EGFR mutation. All factors that were found significant in the univariate logistic regressions were included in the multivariable logistic regression. Odds ratios and 95% confidence intervals were calculated along with likelihood ratio tests for associated *p*-values. We utilized the Kaplan-Meier product limit method to calculate overall survival (OS) stratified by EGFR status at 12, 24 and 36 months. The log-rank test was used to compare the survival functions. All tests were two-sided with statistical significance at a *p*-value < 0.05. All statistical analysis was performed using SAS 9.3 (SAS Institute, Cary, NC, USA). The Northern California Kaiser Foundation Research Institute’s Institutional Review Board approved this study with waiver of consent. 

## 3. Results

### 3.1. Incidence of EGFR Mutation by Clinical Characteristics

We reviewed 293 cases from our regional laboratory database that recorded the testing of EGFR mutations from 2004 to 2012. We excluded 16 cases who had stage I–IIIA disease without evidence of relapse from the analysis. We determined that 277 cases met the inclusion criteria: metastatic (stage IV) NSCLC at the time of diagnosis or relapsed disease after initial definitive treatment (surgery, chemotherapy and radiation). Most cases were metastatic at presentation (91%) and diagnosed between 2008 and 2012 (95%).

Among the 277 cases, 194 tested negative and 83 tested positive for EGFR mutations. The difference in distributions of age at stage IV diagnosis for patients without EGFR mutations when compared to patients with EGFR mutation approaches statistical significance (Median 69 vs. 64, *p* = 0.07, Table 1). A larger proportion of female patients had an EGFR mutation compared to males, but this was also not statistically significant (Table 1). 

The Asian population showed a significantly higher incidence of EGFR mutation on univariate analysis (51.2% versus 20.5% for White and 20.0% for Black, Table 1). In addition, smoking history had a significant impact on the incidence of EGFR mutation in both univariate analysis (*p* < 0.0001, Table 1), as ever-smokers had a 12.2% incidence of EGFR mutation compared to 54.7% for the never-smokers. Among the histologic types, adenocarcinoma was associated with a significantly higher incidence of EGFR mutation on univariate analysis (*p* = 0.02, Table 1). One out of four cases of adenosquamous carcinoma were positive for EGFR mutation (Table 1); this patient tested positive for exon 21 mutation (L858R) and was given 150 mg erlotinib for only 50 days, experienced initial symptomatic improvement but then rapidly deteriorated, and was enrolled into hospice care and expired shortly thereafter. 

We analyzed the association between EGFR mutation and histologic grade and found that patients with grade I and II NSCLC were three-times more likely to harbor EGFR mutations compared to that of grade II–III and III. For the patients with grade I and II, the incidence of EGFR mutation was 38.6% and 36.6% respectively, compared to 6.7% for the patients with grade II–III and 11.3% for grade III histology (*p* < 0.0001, Table 1). 

In multivariable analysis, the difference of incidence of EGFR mutation between sex remained statistically not significant (Table 2). The Asian decent population showed significant higher incidence of EGFR mutation by crude and adjusted analysis (*p* < 0.01 and *p* = 0.05, Table 2), with crude odds ratio (OR) of 4.1 and adjusted OR of 3.4 (both *p* < 0.01, Table 2). There were some differences in the incidence of EGFR mutation among different age groups but this was not statistically significant (Appendix A). The majority of the EGFR mutations were exon 19 deletions and exon 21 L858R point mutations, and there was no significant difference across grades related to the specific mutations (Appendix A). 

In multivariable analysis, compared to other histologies, adenocarcinoma was associated with borderline higher incidence of EGFR mutation, with crude OR of 3.8 (*p* = 0.02) while adjusted OR of 4.9 but only approaching statistical significance (*p* = 0.06, Table 2). The association of grade and the incidence of EGFR mutation remains highly significant in multivariable analysis, with crude OR of 5.0 and adjusted OR of 4.7 (both *p* = 0.01, Table 2). No histologic grade was reported in 15 cases that were positive for EGFR mutations; all these cases were diagnosed based on cytology from pleural fluid. Two out of 21 cases (9.5%) with metastatic squamous cell lung carcinoma tested positive for EGFR exon 19 deletion. One was an ever-smoker and one a never-smoker; however, the number of cases is too small to make a conclusion. 

### 3.2. Histologic Grade Is a Predictive Factor for the Incidence of EGFR Mutation

Asian descent, female sex, adenocarcinoma histology and never-smoking status are established clinical characteristics that are associated with higher incidence of EGFR mutation, compared to non-Asian descent, male sex, squamous cell carcinoma and ever-smokers. To determine if histologic grade can be an additional predictive factor for patients with metastatic lung adenocarcinoma, we analyzed the incidence of EGFR mutations against these established clinical characteristics. We first looked at the ethnicity and then further defined the mutation incidence by histologic grade. As shown in Table 3, for the patients of Asian descent with grade I and II lung adenocarcinoma, the incidence of EGFR mutations was 60.9%, compared to 26.9% for the patients of Asian descent with grade II–III and III histology, a more than two-fold difference (*p* = 0.006). For the non-Asian population (White and Black) with grade I and II histology, the incidence of EGFR mutation was 28.0%, compared to 7.8% for grade II–III to III, a more than three-fold difference (*p* = 0.001). These data indicate that histologic grade is a valuable clinical factor for estimating the incidence of EGFR mutation in both Asian and non-Asian populations with metastatic lung adenocarcinoma.

We further tested the value of histologic grade as a predictive factor of the incidence of EGFR mutations in female patients with lung adenocarcinoma. As shown in the left upper panel of Table 4, the incidence of EGFR mutations in female patients with grade I and II histology was 60.9% for Asian descent, compared to 30.9% for non-Asian descent (*p* = 0.01). The incidence of EGFR mutations in female patients with grade II–III and III metastatic lung adenocarcinoma was 31.3% for Asian descent, compared to 8.3% for non-Asian descent (*p* = 0.09, right upper panel, Table 4, Fishers exact test). This result indicates that histologic grade is associated with the incidence of EGFR mutations in both Asian and non-Asian female patients with metastatic lung adenocarcinoma. 

We also looked at the male patients of Asian and non-Asian descent for incidence of EGFR mutations in relation to histologic grade (Table 4, lower panel). For male patients with grade I and II metastatic lung adenocarcinoma, the incidence of EGFR mutation was 60.9% for Asian descents while 23.7% for non-Asian descents (*p* = 0.004). For male patients with grade II–III and III lung adenocarcinoma, although there was a numerical difference in the incidence of EGFR mutations (20.0% for Asians and 7.1% for non-Asians), the number of patients was small and we did not detect a statistical significance (*p* = 0.28). 

Smoking history is the most significant factor predictive of incidence of EGFR mutations. We analyzed the value of histologic grade against smoking status in our cohort to determine the performance of grade as a predictive factor in estimating the incidence of EGFR mutations. As shown in Table 5, in never-smokers, the incidence of EGFR mutations for patients with grade I and II metastatic lung adenocarcinoma was 59.6%, compared to 38.1% for patients with grade II–III and III histology (*p* = 0.10). In ever-smokers, the incidence of EGFR mutation for patients with grade I and II metastatic lung adenocarcinoma was 16.9%, compared to only 1.8% (1 out of 56 cases, an exon 19 deletion) for patients with grade II–III and III histology (*p* = 0.005). Although the case volume was small in patients whose smoking history was unknown, histologic grade still performed well enough to be suggestive of statistical significance in differentiating the incidence of EGFR mutations (*p* = 0.07).

### 3.3. Treatment with Erlotinib and Overall Survival

To further determine the validity of the cohort of patients in our study, we looked at the OS at 12, 24 and 36 months of patients with and without EGFR mutation. We also examined their treatment history with an EGFR inhibitor. As shown in Table 6, 94% of patients with positive EGFR mutations received erlotinib with a median length of 11 months, versus 21.4% of patients with negative EGFR mutations who received erlotinib with a median length of one month. Five patients in our cohort whose tumor harbored EGFR exon 20 mutation (four with duplication and one insertion) all showed no response to erlotinib. One patient with unknown histologic grade whose tumor harbored an exon 18 mutation G719A and exon 20 mutation R776H also did not respond to erlotinib. The 12-, 24-, and 36-month OS for the patients with EGFR mutations were significantly higher than those for patients without EGFR mutations (*p* < 0.0001, Table 6). Kaplan–Meier estimates of OS show a clear separation of the survival curves, with a median OS of approximately 37 months for patients with EGFR mutations and 13 months for patients without EGFR mutations (Figure 1). These results validate that our cohort of patients is indeed representative of the metastatic lung adenocarcinoma patients in the larger community.

## 4. Discussion

Erlotinib was first approved after the pivotal trial that showed improved OS regardless of EGFR mutational status compared to the best supportive care [10]. The Iressa Pan-Asia Study (IPASS) and the European erlotinib versus chemotherapy trial (EURTAC) showed improved progression-free survival (PFS) when gefitinib or erlotinib was given as first-line therapy for EGFR mutation-positive patients with metastatic NSCLC compared to first-line standard chemotherapy [9,17,18]. The IPASS trial also showed that chemotherapy resulted in superior PFS over gefitinib for EGFR mutation-negative patients [9,16]. These trials led to the recommendation by the NCCN guidelines for testing all patients with metastatic NSCLC before initiating first-line therapy [12]. For a patient whose tumor is positive for an EGFR mutation, erlotinib, afatinib or gefitinib is recommended as the first-line therapy. The quality of life with an oral TKI was shown to be superior to that of first-line chemotherapy for patients with an EGFR mutation [19]. For patients with negative EGFR mutations (and negative for anaplastic lymphoma kinase (ALK) and ROS1 fusion), systemic chemotherapy was the preferred first-line therapy, however, recently two clinical trials have broadened the options. A phase III trial showed that for patients with tumor expressing programmed death ligand 1 (PD-L1) greater than 50%, pembrolizumab resulted in a high response rate, superior PFS and OS as first-line therapy compared to chemotherapy [20]. A phase II trial showed that combination of pembrolizumab with carboplatin and pemetrexed resulted in a higher response rate as first-line therapy regardless of PD-L1 expression, compared to chemotherapy alone [21]. 

The prevalence of EGFR mutations in metastatic NSCLC is less than 15% in the North American population [3,22]. Clinical characteristics including histologic type, smoking status, sex and ethnicity are predictive factors in estimating the probability of harboring an EGFR-activating mutation [3,23]. Our data demonstrate that histologic grade is an additional valuable clinical factor in estimating the incidence of EGFR mutation in both Asian and non-Asian populations, in both females and males and in patients with or without a history of smoking. Our data suggest that for patients with grade II–III or III lung adenocarcinoma, especially those with a history of smoking and high tumor burden with significant symptoms, waiting for the result of EGFR mutational analysis before initiating first-line therapy may not be the best decision because the incidence of positive EGFR mutation is extremely low at 1.8% (one out of 56 cases) for these patients. Promptly initiating systemic chemotherapy or immuno-chemotherapy may be the more appropriate decision, especially if the patient is symptomatic. Non-Asian patients with grade II–III or III lung adenocarcinoma have a less than 10% incidence of EGFR mutation and may also be appropriate for initiating systemic chemotherapy or immuno-chemotherapy promptly while awaiting the result of EGFR mutational analysis. These patients can be switched to erlotinib, afatinib or gefinitib in the event that an EGFR mutation is later detected. Patients with grade II–III or III lung adenocarcinoma can often progress rapidly with potentially irreversible complications if first-line therapy is not promptly initiated [24,25]. For a patient whose histologic grade is I or II, waiting for the result of EGFR mutational analysis is reasonable since the incidence of a EGFR mutation is high and their disease is often less aggressive. 

The definition of histologic grade was based on the 2004 WHO lung cancer classification [15,16] because our study population was from the period of 2004–2012. The 2015 WHO lung cancer classification made modifications on histologic types, however, without significant changes in the definition of histologic grade [26,27]. For example, lung adenocarcinoma with lepidic component is nearly always well- to moderately differentiated and correlates with higher incidence of EGFR mutation [26,28]. The mechanism of higher incidence of EGFR mutation in well- to moderately differentiated lung adenocarcinoma is likely a reflection of lower mutation burden associated with more indolent clinical course driven by the classic EGFR-driver mutation as a characteristic of molecular evolution of cancer initiated by a single driver mutation [28,29]. 

Our study is the first to show that histologic grade is associated with EGFR mutation regardless of ethnicity, sex or smoking status in metastatic lung adenocarcinoma. This finding is of importance in routine clinical practice as using histologic grade to predict the incidence of EGFR mutation is simple and practical. The cohort of patients in our study shares the characteristics reported in the literature correlating the incidence of EGFR mutations with ethnicity, histologic type and smoking status [3,23]. Our data showed a significant difference in OS between the EGFR mutation-positive and -negative populations with differential proportion and length of exposure to erlotinib. This is consistent with the findings from the randomized phase III trials [30,31], though a meta-analysis of 23 heterogeneous trials showed improvement of PFS, but not OS, with TKIs in patients with EGFR mutation-positive metastatic NSCLC [32]. For early-stage NSCLC, a report of a French cohort of 38 cases showed that EGFR mutation was more frequently found in intermediate grade disease [33]. However, in a separate study of early-stage NSCLC, EGFR mutation was found to be of no prognostic value [34]. 

One case of metastatic adenosquamous cell carcinoma (out of four cases) had an exon 19 deletion in our cohort but did not benefit from erlotinib, in contrast to the two patients with metastatic squamous cell carcinoma carrying exon 19 deletion. A previous study from Japan showed that approximately 21.9% out of 32 cases of adenosquamous cell carcinoma of the lung were positive for EGFR mutation [35]. A case report showed a patient with adenosquamous carcinoma and with exon 19 deletion who responded to gefitinib with progression-free survival (PFS) of 19 months [36]. There is evidence that not all EGFR TKIs have the same activity for all EGFR sensitive mutations and the same mutation in different histology may confer differential sensitivity to TKIs [37,38,39,40].

Our study has several limitations, including that it is a retrospective study and that the sample size of the EGFR mutation-positive patients was not sufficiently large to provide the statistical power required in some subset analysis. Additional studies with a larger cohort of EGFR-positive patients in the future shall provide further insight as to what type of EGFR mutations may be associated with histologic grade. For example, is high grade histology associated with the presence of de novo EGFR T790M mutation and the other EGFR mutations associated with resistance to EGFR TKI? The detection of de novo EGFR T790M in metastatic NSCLC was found to be associated with shorter PFS and OS [41,42,43]. 

Our study has several strengths including that the study population was derived from a large organization with more than four million members of diverse ethnicity and demonstrated clinical characteristics representative of similar patients in the larger community; the median follow-up of patients was more than 36 months with clear demonstration of OS difference between the EGFR mutation-positive and -negative populations coupled with the data on the exposure to erlotinib. 

## 5. Conclusions

In summary, histologic grade is a valuable clinical factor predictive of EGFR mutations in metastatic lung adenocarcinoma and may help guide patient selection for promptly initiating systemic therapy in the first-line setting. For this reason, reporting histologic grade for metastatic NSCLC should be a routine practice. 

## Figures and Tables

**Figure 1 medsci-05-00034-f001:**
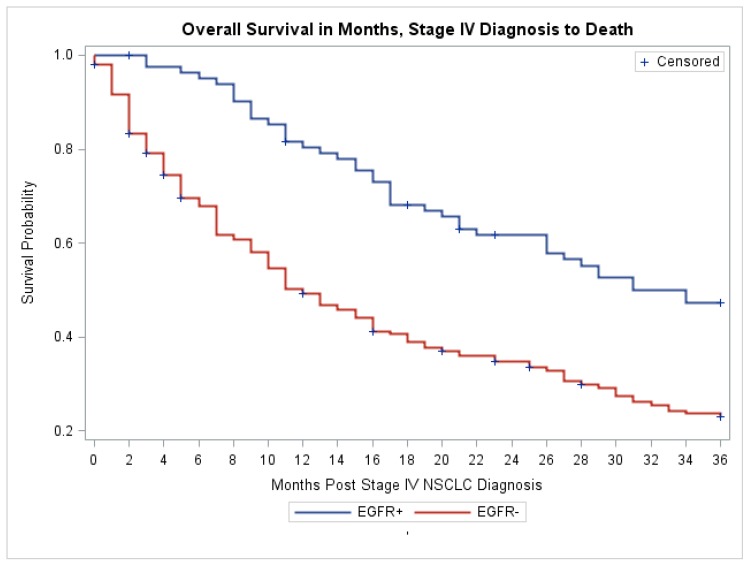
Kaplan–Meier curve of overall survival of patients with negative (Orange, EGFR−) and positive (Blue, EGFR+) EGFR mutations. The numbers in orange and blue represent the number of patients at risk respectively. The numbers in green represent the numbers of months from the diagnosis of stage IV NSCLC to death. NSCLC, non-small cell lung cancer.

**Table 1 medsci-05-00034-t001:** Incidence of epidermal growth factor receptor (EGFR) mutation by clinical characteristics.

Clinical Characteristics	EGFR Mutation Negative N = 194	EGFR Mutation Positive N = 83	Total N = 277 (%)	*p* Value
**Age Median (IQR)**	69.0 (61.0, 76.0)	64.0 (55.0, 75.0)	67.0 (60.0, 76.0)	0.07 *
**Gender N (%)**				
**Male**	89 (73.5)	32 (26.5)	121 (43.7)	0.26
**Female**	105 (67.3)	51 (32.7)	156 (56.3)	
**Race/Ethnicity N (%)**				
**Asian**	42 (48.8)	44 (51.2)	86 (31.1)	<0.0001
**White**	140 (79.6)	36 (20.5)	176 (63.5)	
**Black**	12 (80.0)	3 (20.0)	15 (5.4)	
**History of Smoking N (%)**				
**Ever-smoker**	130 (87.8)	18 (12.2)	148 (53.4)	<0.0001
**Never-smoker**	39 (45.4)	47 (54.7)	86 (31)	
**Unknown**	25 (58.1)	18 (41.9)	43 (15.5)	
**Histologic Type N (%)**				
**Adenocarcinoma**	163 (67.4)	79 (32.6)	242 (87.4)	0.07 **
**Squamous**	19 (90.5)	2 (9.5)	21 (7.6)	
**Adenosquamous**	3 (75.0)	1 (25.0)	4 (1.4)	
**Pure lepedic pattern**	3 (75.0)	1 (25.0)	4 (1.4)	
**Unknown**	6 (100)	0 (0.0)	6 (2.2)	
**Histologic Grade**				
**Grade I**	35 (61.4)	22 (38.6)	57 (20.6)	<0.0001
**Grade II**	59 (63.4)	34 (36.6)	93 (33.6)	
**Grade II–III**	14 (93.3)	1 (6.7)	15 (5.4)	
**Grade III**	86 (88.7)	11 (11.3)	97 (35)	
**Unknown**	0 (0)	15 (100)	15 (5.4)	

All tests are Chi-Square unless otherwise noted. * Mann-Whitney U-test ** Fishers Exact Test. IQR: Interquartile Range.

**Table 2 medsci-05-00034-t002:** Logistic regression of variables predicting of positive EGFR mutation.

Clinical Characteristics	Crude OR (95% CI)	*p*-Value	Adjusted OR (95% CI)	*p*-Value
**Patient Sex**				
**Male**	REF			
**Female**	1.4 (0.8, 2.3)	0.26	--	--
**Age at Stage IV diagnosis**				
**Increase of 5 years**	0.9 (0.8, 1.0)	0.08	--	--
**Race**				
**White**	REF			
**Asian/ Pacific Islander**	4.1 (2.3, 7.1)	<0.01	3.4 (1.7, 6.9)	0.05
**Black**	1.0 (0.3, 3.6)	0.27	1.5 (0.3, 8.6)	0.82
**Smoking Status**				
**Never smoked**	8.7 (4.6, 17.0)	<0.01	8.1 (3.7, 17.5)	<0.01
**Ever smoked**	REF			
**Histology**				
**Adenocarcinoma**	3.8 (1.3, 11.0)	0.02	4.9 (1.0, 24.8)	0.06
**Other**	REF			
**Histology Grade**				
**I and II**	5.0 (2.5, 9.8)	<0.01	4.7 (2.1, 10.5)	<0.01
**II–III and III**	REF			

All tests are Wald Chi-Square. CI: confidence interval; OR: odds ratio. REF: Reference cohort.

**Table 3 medsci-05-00034-t003:** Incidence of EGFR mutation by histologic grade and ethnicity in metastatic lung adenocarcinoma.

Ethnicity	Asian			Non-Asian			
Grade	EGFR Mutation Negative	EGFR Mutation Positive	*p* Value	Grade	EGFR Mutation Negative	EGFR Mutation Positive	*p* Value
**I and II** **N = 46 (%)**	18 (39.1)	28 (60.9)	0.006	**I and II** **N = 93 (%)**	67 (72.0)	26 (28.0)	0.002
**II–III and III** **N = 26 (%)**	19 (73.1)	7 (26.9)		**II–III and III** **N = 64 (%)**	59 (92.2)	5 (7.8)	

All tests are Chi-Square.

**Table 4 medsci-05-00034-t004:** Incidence of EGFR mutation by histologic grade and ethnicity in female and male patient with metastatic lung adenocarcinoma.

Grades I and II				Grades II–III and III			
**Female**	**EGFR Mutation Negative**	**EGFR Mutation Positive**	***p* Value**	**Female**	**EGFR Mutation Negative**	**EGFR Mutation Positive**	***p* Value**
**Asian** **N = 23 (%)**	9 (39.1)	14 (60.9)	0.01	**Asian** **N = 16 (%)**	11 (66.8)	5 (31.3)	0.09 *
**Non-Asian** **N = 55 (%)**	38 (69.1)	17 (30.9)		**Non-Asian** **N = 36 (%)**	33 (91.7)	3 (8.3)	
**Male**	**Grades I and II**		***p* Value**	**Male**	**Grades II–III and III**		***p* Value**
**Asian** **N = 23 (%)**	9 (39.1)	14 (60.9)	0.004	**Asian** **N = 10 (%)**	8 (80.0)	2 (20.0)	0.28 *
**Non-Asian** **N = 38 (%)**	29 (76.3)	9 (23.7)		**Non-Asian** **N = 28 (%)**	26 (92.9)	2 (7.1)	

All tests are Chi-Square unless otherwise noted. * Fishers exact test.

**Table 5 medsci-05-00034-t005:** Incidence of EGFR mutation by histologic grade and by smoking status in patients with metastatic lung adenocarcinoma.

Never-Smokers				Ever-Smokers				Smoking History Unknown			
Grade	EGFR Mutation Negative	EGFR Mutation Positive	*p* Value	Grade	EGFR Mutation Negative	EGFR Mutation Positive	*p* Value	Grade	EGFR Mutation Negative	EGFR Mutation Positive	*p* Value
**I and II** **N = 52 (%)**	21 (40.4)	31 (59.6)	0.10 *	**I and II** **N = 65 (%)**	54 (83.1)	11 (16.9)	0.005 **	**I and II** **N = 22 (%)**	10 (45.5)	12 (54.5)	0.07 **
**II–III and III** **N = 21 (%)**	13 (61.9)	8 (38.1)		**II–III and III** **N = 56 (%)**	55 (98.2)	1 (1.8)		**II–III and III** **N = 13 (%)**	10 (76.9)	3 (23.1)	

* Chi-Square Test ** Fishers exact test.

**Table 6 medsci-05-00034-t006:** Length of treatment with erlotinib and overall survival for patients with positive or negative EGFR mutation.

	EGFR Mutation Positive N = 83	EGFR Mutation Negative N = 194	*p* Value
Number of patients received erlotinib	78 (94%)	42 (21.6%)	<0.0001 **
Median length of exposure (month) [Quartile1, Quartile 3]	11 [5, 33]	1 [0, 5]	<0.0001 **
Overall Survival			
12-month	81.65%	49.04%	<0.0001 *
24-month	61.67%	34.04%	
36-month	47.24%	22.96%	

* Mann-Whitney-U test ** Log-rank test.

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
