# Peer review of "Histologic Grade Is Predictive of Incidence of Epidermal Growth Factor Receptor Mutations in Metastatic Lung Adenocarcinoma"

_medsci, 2017, doi:10.3390/medsci5040034_

Reviewer 1 Report

The manuscript by Levy et al. entitled “Histologic grade is predictive of incidence of epidermal growth factor receptor mutation in metastatic lung adenocarcinoma” is an attempt to demonstrate a correlation between histological grades and EGFR mutational status; the underlying assumption being that these grades may help in therapeutic decision making. The article is overall well written but comes with all the limitations of a retrospective study that uses clinical data for research purposes. Nonetheless, the manuscript retains its value. I have the following remarks:

1. Please provide the IRB number of your study in the materials and methods section. It has been shown but at the end of the document;

2. What is the average waiting time for EGFR mutational analysis that may justify the use of histological grade in clinical decision instead?

3. Lanes 93 – 95: “This section may be divided by subheadings. It should provide a concise and precise description of the experimental results, their interpretation as well as the experimental conclusions that can be drawn”. It seems like this is an instruction for the authors but not really part of the manuscript.

4. The organization of your data is somewhat confusing. It is common to provide the univariate analyses, followed by the multiple regression with in your case is a binary logistic regression. It turns out that the logistic regression is displayed early in Table 2, which gives the feeling of going back and forth when reading your manuscript.

5. Table 4 should be improved (the non-asian row is not well displayed)

6. Please provide all statistical methods used under all tables.

7. Page 8: the kaplan meier curve should be improved. It is usual to display the legend within the figure (EGFR+/EGFR-). No need to show the raw number in the x axis.

8. It will be interesting to discuss this finding that activating mutations of EGFR are rather common in low grades NSCLC. What, according to the authors' opinion, may explain this fact? 

9.  Typos: lane 216 for patients (s)

Author Response

The manuscript by Levy et al. entitled “Histologic grade is predictive of incidence of epidermal growth factor receptor mutation in metastatic lung adenocarcinoma” is an attempt to demonstrate a correlation between histological grades and EGFR mutational status; the underlying assumption being that these grades may help in therapeutic decision making. The article is overall well written but comes with all the limitations of a retrospective study that uses clinical data for research purposes. Nonetheless, the manuscript retains its value. I have the following remarks:

Response: Thank you for your excellent comment and suggestions, we have made revisions accordingly below.

1. Please provide the IRB number of your study in the materials and methods section. It has been shown but at the end of the document;

 Response: This is corrected now.

2. What is the average waiting time for EGFR mutational analysis that may justify the use of histological grade in clinical decision instead?

Response: We believe the average wait-time should be less than 10 days. However, the wait-time is significantly longer than 10 days in most practices in the world.

3. Lanes 93 – 95: “This section may be divided by subheadings. It should provide a concise and precise description of the experimental results, their interpretation as well as the experimental conclusions that can be drawn”. It seems like this is an instruction for the authors but not really part of the manuscript.

Response: this paragraph was inadvertently left here when the manuscript was reformatted. We have removed it now.  

4. The organization of your data is somewhat confusing. It is common to provide the univariate analyses, followed by the multiple regression with in your case is a binary logistic regression. It turns out that the logistic regression is displayed early in Table 2, which gives the feeling of going back and forth when reading your manuscript.

Response: we have made changes in the Result section to avoid the back-and-forth data presentation to eliminate the confusion, including separating the data presentation of Table 1 and Table 2 into separate paragraphs.  

5. Table 4 should be improved (the non-asian row is not well displayed:

Response: The display is corrected now.  

6. Please provide all statistical methods used under all tables.

Response: We have provided statistical methods under all the tables now.  

7. Page 8: the kaplan meier curve should be improved. It is usual to display the legend within the figure (EGFR+/EGFR-). No need to show the raw number in the x axis.

Response: We have removed the raw numbers in the figure as suggested.  

8. It will be interesting to discuss this finding that activating mutations of EGFR are rather common in low grades NSCLC. What, according to the authors' opinion, may explain this fact? 

Response: This is an excellent suggestion and we have added discussion (line 294 to 297).  

9. Typos: lane 216 for patients (s)

Response: This is corrected now.

Reviewer 2 Report

This is an interesting and important study. The authors examined 277 patients with metastatic NSCLC and identified that histologic grade is a predictive factor for the incidence of EGFR mutation independent of gender, ethnicity, and smoking status. This reviewer raises following issues.

- The authors used the previous 2004 WHO criteria of lung cancer. In the current 2015 WHO criteria, the definition of adenocarcinoma and squamous cell carcinoma differed as well summarized in a review article (https://www.ncbi.nlm.nih.gov/pubmed/28894699). This modification is so important that the author should address this. This modification can possibly affect the results of this study; therefore, this can be one of limitations.

- Positivity of EGFR mutation is associated with lepidic component, a micropapillary pattern, and the hobnail cell type, as well summarized in a review article (https://www.ncbi.nlm.nih.gov/pubmed/20073607). This morphologic feature can be a potential confounder for the assessment of predictive factor for the incidence of EGFR mutation. The authors should address this. Nontheless, using the histologic grade is simple and more practical than using the characteristic morphological features. The authors should make a point on the usefulness to simplicity of using the histologic grade.

Author Response

Comments and Suggestions for Authors

This is an interesting and important study. The authors examined 277 patients with metastatic NSCLC and identified that histologic grade is a predictive factor for the incidence of EGFR mutation independent of gender, ethnicity, and smoking status. This reviewer raises following issues.

Response: Thank you for your excellent comment and suggestions. We have made revisions accordingly.

- The authors used the previous 2004 WHO criteria of lung cancer. In the current 2015 WHO criteria, the definition of adenocarcinoma and squamous cell carcinoma differed as well summarized in a review article (https://www.ncbi.nlm.nih.gov/pubmed/28894699). This modification is so important that the author should address this. This modification can possibly affect the results of this study; therefore, this can be one of limitations.

Response: This is an excellent suggestion. We have added a paragraph to address this question, (line 289 to line 297). Also, we have made improvement in the Introduction section with additional background information and two added references, ref.13 and 14).

- Positivity of EGFR mutation is associated with lepidic component, a micropapillary pattern, and the hobnail cell type, as well summarized in a review article (https://www.ncbi.nlm.nih.gov/pubmed/20073607). This morphologic feature can be a potential confounder for the assessment of predictive factor for the incidence of EGFR mutation. The authors should address this. Nontheless, using the histologic grade is simple and more practical than using the characteristic morphological features. The authors should make a point on the usefulness to simplicity of using the histologic grade.

Response: We completely agree with reviewer’s comment, and thank you for another great suggestion. We have added a paragraph (line 289 to line 297) for further discussion of histologic types and grade with 4 additional references (ref. 26-29) as suggested, and a sentence (line 300 to 301) that emphasizes the simplicity and practicality of using grade as a predictive tool.

Round  2

Reviewer 2 Report

The authors adequately revised the manuscript.